

# Dynamics of Zika virus outbreaks: an overview of mathematical modeling approaches

Anuwat Wiratsudakul[1,2], Parinya Suparit[3] and Charin Modchang[3,4]

[1] Department of Clinical Sciences and Public Health, Faculty of Veterinary Science, Mahidol University, Phutthamonthon, Nakhon Pathom, Thailand
[2] The Monitoring and Surveillance Center for Zoonotic Diseases in Wildlife and Exotic Animals, Faculty of Veterinary Science, Mahidol University, Phutthamonthon, Nakhon Pathom, Thailand
[3] Biophysics Group, Department of Physics, Faculty of Science, Mahidol University, Ratchathewi, Bangkok, Thailand
[4] Centre of Excellence in Mathematics, CHE, Ratchathewi, Bangkok, Thailand

## ABSTRACT

**Background**. The Zika virus was first discovered in 1947. It was neglected until a major outbreak occurred on Yap Island, Micronesia, in 2007. Teratogenic effects resulting in microcephaly in newborn infants is the greatest public health threat. In 2016, the Zika virus epidemic was declared as a Public Health Emergency of International Concern (PHEIC). Consequently, mathematical models were constructed to explicitly elucidate related transmission dynamics.

**Survey Methodology**. In this review article, two steps of journal article searching were performed. First, we attempted to identify mathematical models previously applied to the study of vector-borne diseases using the search terms "dynamics," "mathematical model," "modeling," and "vector-borne" together with the names of vector-borne diseases including chikungunya, dengue, malaria, West Nile, and Zika. Then the identified types of model were further investigated. Second, we narrowed down our survey to focus on only Zika virus research. The terms we searched for were "compartmental," "spatial," "metapopulation," "network," "individual-based," "agent-based" AND "Zika." All relevant studies were included regardless of the year of publication. We have collected research articles that were published before August 2017 based on our search criteria. In this publication survey, we explored the Google Scholar and PubMed databases.

**Results**. We found five basic model architectures previously applied to vector-borne virus studies, particularly in Zika virus simulations. These include compartmental, spatial, metapopulation, network, and individual-based models. We found that Zika models carried out for early epidemics were mostly fit into compartmental structures and were less complicated compared to the more recent ones. Simple models are still commonly used for the timely assessment of epidemics. Nevertheless, due to the availability of large-scale real-world data and computational power, recently there has been growing interest in more complex modeling frameworks.

**Discussion**. Mathematical models are employed to explore and predict how an infectious disease spreads in the real world, evaluate the disease importation risk, and assess the effectiveness of intervention strategies. As the trends in modeling of infectious diseases have been shifting towards data-driven approaches, simple and

Corresponding author
Charin Modchang,
charin.mod@mahidol.edu

complex models should be exploited differently. Simple models can be produced in a timely fashion to provide an estimation of the possible impacts. In contrast, complex models integrating real-world data require more time to develop but are far more realistic. The preparation of complicated modeling frameworks prior to the outbreaks is recommended, including the case of future Zika epidemic preparation.

## INTRODUCTION

Zika is a single-stranded RNA flavivirus, a member of the *Flaviviridae* family (*Lopes, Miyaji & Infante, 2016*). The virus is genetically related to some others responsible for encephalitis in humans, including chikungunya, dengue, Japanese encephalitis, West Nile, and the yellow fever virus (*Lucey & Gostin, 2016*; *Goeijenbier et al., 2016*; *Vest, 2016*). Zika is one of the arboviruses transmitted by *Aedes* mosquitoes. The main vectors are *Aedes aegypti* and *Aedes albopictus* (*Al-Qahtani et al., 2016*). These mosquitoes are mostly found in tropical and subtropical regions (*Petersen et al., 2016*).

The Zika virus was first discovered in rhesus monkeys in 1947 while researchers were studying yellow fever in Zika Forest, Uganda, and it was isolated from *Aedes africanus* mosquitoes the subsequent year (*Dick, Kitchen & Haddow, 1952*). The first human isolation was recorded in Nigeria six years later (*MacNamara, 1954*; *Petersen et al., 2016*). For decades, the viral infection was sporadically reported in Africa and Southeast Asia (*Hayes, 2009*; *Goeijenbier et al., 2016*). The first large outbreaks occurred on Yap Island, Federated States of Micronesia, in 2007 (*Duffy et al., 2009*). In this epidemic, 49 confirmed cases were found together with another 59 probable cases. It was estimated that up to 73% of the Yap Island residents were asymptomatically infected (*Duffy et al., 2009*; *Kindhauser et al., 2016*). The episodes of large-scale Zika virus outbreaks happened in 2013, when the virus migrated to French Polynesia, a French territory located in the South Pacific. This outbreak was the largest recorded at the time (*Cao-Lormeau et al., 2014*; *Cao-Lormeau et al., 2016*; *Musso, 2015*). Overall, 19,000 suspected cases were estimated throughout the epidemic's course (*Cao-Lormeau et al., 2014*). The first evidence of Guillain-Barré syndrome related to the Zika virus was also seen in this historic outbreak (*Cao-Lormeau et al., 2016*). Subsequently, the virus from French Polynesia dispersed to many countries in the Pacific Ocean, finally reaching Easter Island, Chile, in 2014 (*Tognarelli et al., 2015*). The virus seems to have established well on the continent, especially in Latin American countries (*Shi et al., 2016*). For example, the autochthonous transmission was first confirmed in Brazil in 2015 (*Zanluca et al., 2015*) and the Brazilian Ministry of Health estimated the number of suspected cases at 440,000 to 1,300,000 that year. The Zika infection was also linked to the unusual rising incidence of microcephaly in newborn infants (*Mlakar et al., 2016*; *De Oliveira & Da Costa Vasconcelos, 2016*; *Heymann et al., 2016*) together with some other neurological disorders

including Guillain-Barré syndrome (*De Oliveira et al., 2017*). On February 1, 2016, the World Health Organization (WHO) Director-General declared Zika virus outbreaks in Latin American countries as a Public Health Emergency of International Concern (PHEIC) (*Heymann et al., 2016*). As of March 9, 2017, vector-borne Zika virus transmission was found in 84 countries, territories, or subnational areas (*WHO, 2017*).

In addition to Zika, in the twenty-first century many emerging and reemerging infectious diseases threaten the human race. With rapid globalization, these diseases are often disseminated at unprecedented speed. The epidemics of severe acute respiratory syndrome (SARS) in 2003 and the H1N1 influenza pandemic of 2009 are excellent evidence in the first decade (*Mackey & Liang, 2012*). More recently, we face new threats almost every year, for example, the Middle East respiratory syndrome coronavirus (MERS-CoV) in Saudi Arabia in 2012 (*De Groot et al., 2013*), the Ebola virus in the West African region in 2014 (*WHO Ebola Response Team, 2014*). In such epidemics, the real-time evaluation of the ongoing situation is vitally important. To serve this purpose, mathematical modeling has been exploited to monitor the outbreak progression, predict the trend of disease transmission, and tailor related control strategies (*McVernon, McCaw & Mathews, 2007*; *De Jong & Hagenaars, 2009*).

Infectious disease modeling is an interdisciplinary approach. Modelers are obligated to comprehend not only the mathematical frameworks but also the biological knowledge behind the epidemics (*Rock et al., 2014*). Recently, mathematical modeling has been well established as an epidemiological tool. It has been used to combat many infectious diseases. The very first mathematical model was traced back to the work of Daniel Bernoulli in the eighteenth century. Bernoulli employed a simple model to estimate life expectancy due to variolation practices in smallpox epidemics (*Bernoulli, 1766*). However, the modern era of infectious disease modeling was actually initiated a century ago with a mosquito-borne model proposed by Sir Ronald Ross. Ross developed a set of mathematical equations to illustrate how malaria parasites were transmitted between mosquitoes and humans (*Ross, 1911*). The model was later complemented by the work of Macdonald (*MacDonald, 1952*), and finally became the well-known Ross-Macdonald models. This modeling framework still plays an important role in research on malaria and other mosquito-borne diseases (*Smith et al., 2012*). Nevertheless, there were also many other scientists working on malaria transmission dynamics. For instance, Kermack and McKendrick incorporated the law of mass action into the Ross model and proposed new and modern compartmental models (*Kermack & McKendrick, 1927*; *Kermack & McKendrick, 1932*; *Kermack & McKendrick, 1933*) that later became the most widely used basic structures in infectious disease modeling.

The present review aims to provide an overview of mathematical modeling methods, particularly those developed for Zika virus transmission. However, it is not possible to cover, in a review, all kinds of mathematical models applied to infectious disease studies. In this review, we describe some common models developed thus far. We explain different approaches ranging from simple compartments to sophisticated models integrating real-word data. The idea is to provide some basic knowledge of mathematical projections before

further exploring the models, particularly those developed for Zika virus transmission. We also discuss recent advances and trends of research in the infectious disease modeling.

## SURVEY METHODOLOGY

We attempt to cover different types of mathematical models applied to the study of vector-borne disease, particularly the Zika virus. First, we provide basic knowledge on methodological approaches in order to facilitate non-mathematical background readers. We therefore initiated our survey to investigate previously published modeling frameworks. Subsequently, we further explore specifically the use of models in the study of the Zika virus. In our publication survey, we used the Google Scholar (https://scholar.google.com/) and PubMed (http://www.ncbi.nlm.nih.gov/pubmed/) databases to search for the relevant peer-reviewed journal articles. In the first step, we used the search terms "dynamics," "mathematical model," "modeling," and "vector-borne" together with the names of vector-borne diseases including chikungunya, dengue, malaria, West Nile, and Zika. Then, we expanded our search to include the related models identified by the prior screening. The secondary search terminologies included "compartmental," "spatial," "metapopulation," "network," "individual-based," and "agent-based." In the second step, we examined only the models applied to Zika virus simulations. We strictly searched for publications focusing on the applications of mathematical modeling in Zika virus research. The search terms were then designated as the names of the modeling techniques described earlier AND "Zika." We consistently excluded unrelated studies throughout the review process. For the publications that met our criteria, we intensively reviewed their modeling methods, categorized into the modeling types and compared to other related studies we found. The papers with irrelevant methodology were then removed.

As we tried to capture all available studies, the publication year was unrestricted. However, the mathematical modeling approach in the Zika virus study has recently emerged. Hence, most of the research was recently published. We have collected research articles that were published before August 2017 based on our search criteria.

## RESULTS

We found five basic model architectures previously applied to vector-borne research. These include compartmental, spatial, metapopulation, network, and individual-based models. We reviewed these accordingly.

### Basic compartmental model

In the classical compartmental model, the whole population is divided into groups according to individual health status (*Hethcote, 2000*). For example, in the SIR model, the population is split into the compartments of *susceptible* (S, healthy individuals), *infectious* (I, diseased and contagious individuals), and *recovered* (R, immune individuals). During the course of disease transmission, each individual may progress across the compartments,

with the rate illustrated by these ordinary differential equations (Fig. 1A):

$$\frac{dS}{dt} = -\frac{\beta SI}{N},$$

$$\frac{dI}{dt} = \frac{\beta SI}{N} - \gamma I, \quad \text{and} \tag{1}$$

$$\frac{dR}{dt} = \gamma I,$$

where $\beta$ denotes the transmission rate, dictating the speed at which susceptible individuals become infectious, and $\gamma$ represents the recovery rate, which defines how fast the infectious individuals recovered from the disease. The force of infection in this case is defined as $\beta I/N$, where $N$ is the total population. In this simplest case, it is assumed that the dynamics of disease transmission are much faster than the dynamics of demographic processes, for example, births, deaths, and migration; hence, these demographic dynamics can be ignored. In addition to SIR, other forms of compartmental models exist, for instance, SI, SIS, SIRS, SEIR, SEIRS, MSIR, MSEIR, and MSEIRS, among others. E and M are acronyms for *exposed* (individuals already exposed to the disease but not yet infectious) and *maternal* (those with maternal immunity), respectively. The inclusion of different compartments is based on the nature of the diseases (*Hethcote, 2000*). The models have been applied to many emerging infectious diseases, for example, avian influenza (*De Jong & Hagenaars, 2009*), Ebola (*Browne, Gulbudak & Webb, 2015*; *Khan et al., 2015*; *Santermans et al., 2016*; *Asher, 2017*), HIV/AIDS (*Akpa & Oyejola, 2010*; *Luo et al., 2015*), and many others.

One of the most important parameters that is always measured in the compartmental model is the basic reproduction number $R_0$. The $R_0$ is defined as "the average number of secondary cases produced by a single infectious individual in a totally susceptible population in the initial stage of the outbreak" (*Hethcote, 2000*; *Rock et al., 2014*). The $R_0$ is regarded as a threshold at which the epidemic is still progressing. The infection may persist and the transmission continues if the $R_0$ is greater than 1, whereas the epidemic is going to cease in the long term when the $R_0$ is otherwise (*Hethcote, 2000*; *Rock et al., 2014*; *Sidiki & Tchuente, 2014*). This parameter is estimated by $\beta/\gamma$ in the SIR framework (*Rock et al., 2014*). Nonetheless, the $R_0$ varies considerably from disease to disease. For example, the approximate $R_0$ for measles, mumps, and polio is 16, 12, and 5, respectively (*Glomski & Ohanian, 2012*). Furthermore, the $R_0$ values are also different in the same disease but at a different place and time. For example, during the 2014–2015 Ebola virus outbreaks, the $R_0$ values were 1.71 for Guinea, 1.83 for Liberia, and 2.02 for Sierra Leone (*WHO Ebola Response Team, 2014*). Therefore, the $R_0$ is not likely referable across spatiotemporal entities.

## Vector-borne compartmental model

The models applied for vector-borne diseases are still globally based on the standard compartmental model. Nonetheless, the compartments designed to visualize the dynamics of vector populations are always incorporated. Indeed, the vector-borne model accounts for a multi-species approach involving interspecies disease transmission. Hosts and vectors
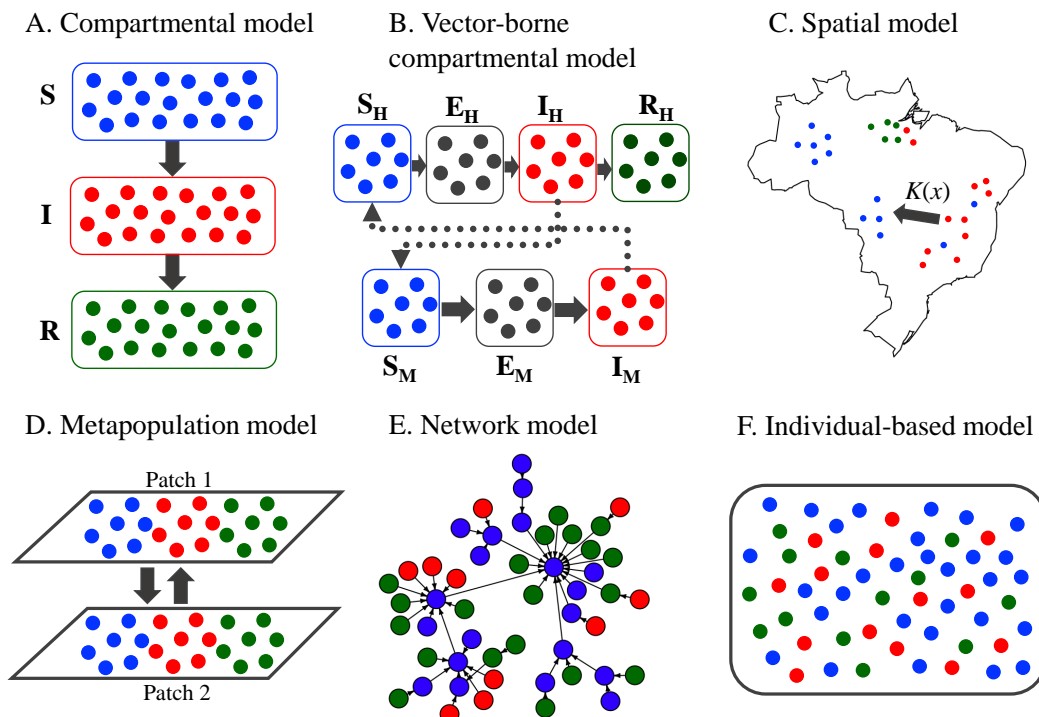

**Figure 1** **Conceptual frameworks of different epidemic models.** The colors represent epidemiological status: susceptible (S, blue), exposed (E, gray), infectious (I, red), and recovered (R, green). (A) Basic SIR compartmental model. Individuals are assumed to be well-mixed and are classified only according to their epidemiological status. (B) Vector-borne compartmental model. The subscripts H and M denote human and mosquito, respectively. Both host and vector individuals are assumed to be well-mixed and are classified only according to their epidemiological status. (C) Spatial model. Individuals are located at different locations. The transmission of infection between an infectious individual and a susceptible individual at distance $x$ may occur with probability $K(x)$. (D) Metapopulation model. The entire population is divided into two distinct subpopulations, each with independent disease transmission dynamics, together with interactions between subpopulations. The subpopulation in each patch is mixed homogeneously. (E) Network model. The model is formed by at least two basic components: vertex and edge. Vertices are connected by edges defined by the relationship of interest such as trade or travel. Infectious diseases are modeled to spread via the edges in this model. (F) Individual-based model. In this most complicated model, the stochastic epidemiological dynamics for each individual can be explicitly simulated with a set of characteristics including epidemiological status, spatial location, interaction preference, behavior traits, etc.

must be present; otherwise, the pathogen cannot spread. The most notable model may refer to the Ross-Macdonald models (*Ross, 1911*; *MacDonald, 1952*) for the transmission of malaria. However, the vector-borne models are often represented in the SEIR and SEI frameworks for human and vector compartments, respectively (Fig. 1B). Here, we demonstrate a model developed to illustrate Zika virus transmission (*Funk et al., 2016*):

**Hosts**

$$\frac{dS_H}{dt} = -\lambda_H S_H,$$

$$\frac{dE_H}{dt} = \lambda_H S_H - \delta_H E_H,$$

$$\frac{dI_H}{dt} = \delta_H E_H - \gamma_H I_H,$$

$$\frac{dR_H}{dt} = \gamma_H I_H, \tag{2}$$

**Mosquitoes**

$$\frac{dS_M}{dt} = \vartheta_M - \lambda_M S_M - \mu_M S_M,$$

$$\frac{dE_M}{dt} = \lambda_M S_M - (\delta_M + \mu_M) E_M, \quad \text{and}$$

$$\frac{dI_M}{dt} = \delta_M E_M - \mu_M I_M,$$

where the subscripts $H$ and $M$ stand for the host and mosquito, respectively. The parameters $\lambda, \delta, \vartheta$, and $\mu$ represent the force of infection, incubation rate, birth rate, and death rate, respectively. The forces of infection for humans and mosquitoes were calculated as:

$$\lambda_H = a p_H m I_M \quad \text{and} \tag{3}$$

$$\lambda_M = a p_M \frac{I_H}{N_H},$$

where $a$ is the mosquito biting rate, $p_H$ is the probability that a bite from an infectious mosquito will lead to infection in human, $p_M$ is the probability of a mosquito being infected from biting an infectious human, $N_H$ is the number of human individuals, and $m$ represents the number of mosquitoes contacting one human.

To get a better understanding of the range of dynamics in these vector-borne diseases, we calculate the number of secondary human cases generated from an average human case, incorporating the cycle of transmission through the vector. To do so, we start with one freshly infected human. From this primary human case, the expected number of infected mosquitoes is the product of the infectious duration in humans, the rate of disease transmission, and the probability that a newly infected mosquito will progress to the infectious stage : $(\frac{1}{\gamma_H})\lambda_M(\frac{\delta_M}{\delta_M + \mu_M})$. Similarly, the expected number of human individuals infected by an infectious mosquito is calculated as $(\frac{1}{\mu_M})\lambda_H N_H$. Thus, the $R_0$ is given by the product of these two terms (*Rock et al., 2014*)

$$R_0 = \left(\frac{1}{\gamma_H}\right)\lambda_M\left(\frac{\delta_M}{\delta_M + \mu_M}\right)\left(\frac{1}{\mu_M}\right)\lambda_H N_H = \frac{a^2 p_M p_H m \delta_M}{\gamma_H \mu_M (\delta_M + \mu_M)}. \tag{4}$$

It is noteworthy that this value of $R_0$ that includes a complete cycle of transmission is the square of the value calculated using the next-generation matrix approach (*Diekmann, Heesterbeek & Roberts, 2010*); however, they agree on the invasion threshold.

Like direct-contact diseases, the $R_0$ for vector-borne epidemics varies across space–time settings. For example, the $R_0$ for the dengue virus in Brazil ranged from 2–103 in different epidemics in the country from 1996–2003 (*Tabachnick, 2016*). For Zika, the $R_0$ in the outbreaks on Yap Island was estimated between 4.3 and 5.8 in 2007, whereas the value was found at 1.8–2.0 in the French Polynesian epidemics in 2013–2014 (*Nishiura et al., 2016a*). In the recent Zika virus epidemic in Columbia in 2015–2016, the $R_0$ was approximately 2.2–14.8 (*Nishiura et al., 2016b*).

## Spatial epidemic model

According to the first law of geography proposed by Waldo Tobler, "everything is related to everything else, but near things are more related than distant things" (*Tobler, 1970*). This idea has become a fundamental concept of spatial studies. Spatial epidemiology is a field concerning the geographical distributions of disease incidences (*Lawson, 2013*). The most primitive tool is disease mapping. However, spatial modeling is much more advanced. This method incorporates the spatial features of disease occurrences and disease transmission behaviors. In many cases, diseases were observed to spread around the index case. One of the best examples is the airborne virus foot-and-mouth disease (FMD). The FMD virus is capable of transmission by air up to 60 km on land and up to 250 km above water bodies (*Lee et al., 2013*). In addition, spatial cluster causes closer places to become more vulnerable (*Lessler et al., 2016a*; *Lessler, 2016b*). To calculate the spatial probability, the transmission kernel is calculated. The transmission kernel is defined as the probability distribution of distances between the infectious premise and other related places (*Lessler et al., 2016a*; *Lessler, 2016b*). The estimation of this parameter can be performed using various forms, for example, exponential (E), Gaussian (G), and fat-tailed (F) methods, which are demonstrated as

$$
\begin{aligned}
K_E(x) &= \alpha e^{-\alpha x}, \\
K_G(x) &= \frac{\alpha}{\sqrt{\pi}} e^{-\alpha^2 x^2}, \quad \text{and} \\
K_F(x) &= \frac{\alpha}{4} e^{-\alpha^{1/2} x^{1/2}},
\end{aligned}
\tag{5}
$$

where $\alpha$ denotes the kernel parameter (*Szmaragd et al., 2009*).

The spatial epidemic models have also been applied to vector-borne diseases. For instance, in the studies of dengue (*Delmelle et al., 2016*; *Sardar & Saha, 2017*; *Vincenti-Gonzalez et al., 2017*), West Nile (*Crowder et al., 2013*; *Harrigan et al., 2014*; *Lin & Zhu, 2017*), and Zika (*Fitzgibbon, Morgan & Webb, 2017*), different modeling approaches were used. In the study of the dengue virus, a power-law form time-dependent transmission kernel (*Sardar & Saha, 2017*), hot-spot detection and risk factor analysis (*Vincenti-Gonzalez et al., 2017*), and geographically weighted regression model (*Delmelle et al., 2016*) were used to illustrate how the virus spreads. In the West Nile virus study, a weighted ensemble model (*Harrigan et al., 2014*) and a spatially explicit model incorporating land-use and climate variables (*Crowder et al., 2013*) as well as a reaction–diffusion model using a spatial–temporal risk index (*Lin & Zhu, 2017*) were constructed to explain the spatial diffusion of the virus under different circumstances. For Zika, spatially dependent differential equations were employed to describe the 2015–2016 Zika outbreak in Rio de Janeiro Municipality in Brazil (*Fitzgibbon, Morgan & Webb, 2017*). A location-specific projection was also performed to estimate the magnitude of Zika virus infections in childbearing women on the American continent (*Perkins et al., 2016*).

## Metapopulation model

The term "metapopulation" was coined by Richard Levins in 1969 (*Levins, 1969*) to systematically delineate the dynamics of insect pest population in farms. However, the

term and its concepts have been widely expanded to different scientific communities including epidemiology. Metapopulation assumes that the whole population is divided into different discrete spatial subgroups called "patches." The subpopulation mixes homogeneously whereas the contact between the patches only occurs at some rates (*Rock et al., 2014*). Consequently, we can fit compartmental models such as SIR and SEIR into each patch to better project how the disease of interest spreads within the subgroups (*Rock et al., 2014*; *Wang & Li, 2014*). At this point, the metapopulation becomes the combination of compartmental and spatial epidemic models. This approach allows us to simulate a very large population with a well-defined spatial distribution (*Banos et al., 2015*).

For a SIR-based metapopulation model, a suitable modified version of the classical SIR approach, Eq. (1), would be

$$\frac{dS_i}{dt} = -\lambda_i S_i,$$

$$\frac{dI_i}{dt} = \lambda_i S_i - \gamma I_i, \quad \text{and} \tag{6}$$

$$\frac{dR_i}{dt} = \gamma I_i,$$

where the subscript $i$ indicates the parameters and variables that are particular to patch $i$. The force of infection, $\lambda_i$, incorporates transmission from both the infectious individuals within patch $i$ and the infectious individuals from patch $j$. The exact formula of $\lambda_i$ depends on the assumed mechanism of transmission and the strength of the interaction between the patches. In general, the force of infection is expressed as (*Rock et al., 2014*; *Sornbundit, Triampo & Modchang, 2017*)

$$\lambda_i = \sum_{j=1}^{n} \beta_{ij} \frac{I_j}{N_j}, \tag{7}$$

where $\beta_{ij}$ is the transmission rate from the infectious individuals in patch $j$ to the susceptible individuals in patch $i$, $N_j$ is the total number of individuals in patch $j$, and $n$ is the number of patches.

In vector-borne disease modeling, the ideas of metapopulation have already been deeply imbedded. The models always involve different subgroups, that is, the hosts and vectors. In many cases, spatial distribution patterns were concurrently considered. As demonstrated in a previous dengue study (*Lee & Castillo-Chavez, 2015*), a two-patch model was constructed to explore the influence of between-patch human movements on viral transmission dynamics. In the patches, the SEIR and SEI models were architected for human and mosquito populations, respectively. This was done to imitate how diseases spread within subpopulations. Another example is a study on the impact of human movement on the dynamics and persistence of vector-borne diseases at the city scale (*Adams & Kapan, 2009*). The authors constructed metapopulation models which assume that human population lives in a home patch free of mosquitoes but moves to and fro patches with immobile mosquito subpopulations. Different human movement patterns were represented by different connection patterns between human and mosquito subpopulations. It was found

that more variable human movement pattern increases the influence of the large vector population patches in establishing new foci of transmission and enhances pathogen persistence (*Adams & Kapan, 2009*). In Zika virus research, a metapopulation-typed model was constructed to investigate the effects of sexual transmission and human migration in the spread of the virus (*Baca-Carrasco & Velasco-Hernández, 2016*). Recently, Zhang et al. expanded the Global Epidemic and Mobility Model (GLEAM) (*Balcan et al., 2010*), a metapopulation model integrating real-world demographic data and human mobility patterns, to incorporate data on mosquito density and entomological-related parameters. The expanded GLEAM model was employed to analyze the spread of the Zika virus in the Americas. It was estimated that the first introduction of the virus to Brazil may have occurred between August 2013 and April 2014 (*Zhang et al., 2017*).

## Network model

In fact, the interactions between actors in mathematical models are governed by the concept of the contact network. It is assumed in the homogeneous compartmental model that all individuals are linked by a regular random pattern (*Bansal, Grenfell & Meyers, 2007*). On the other hand, the heterogeneous models, namely spatial and metapopulation, possess different assumptions that take into account the higher realistic contact structures. The idea of a contact network emerged from the mathematical graph theory and was first used in social sciences. Two fundamental components that form a network are called "vertex" and "edge" (*Lanzas & Chen, 2015*). A vertex is a unit of interest for an individual, a group of people, a village, a city, or even an entire country. An edge is the link between a pair of vertices. The edge represents the bond between vertices, which is important in disease transmission, such as animal movement or human transportation. The interaction is further divided into directed and undirected (*Martínez-López, Perez & Sánchez-Vizcaíno, 2009*), of which the directed links dictate the incoming and outgoing edges; for example, flight itineraries, whereas the undirected approach does not consider directions, such as co-author networks. In epidemiology, contact network modeling has often been used to investigate disease transmission in both humans (*Vazquez-Prokopec et al., 2013*; *Machens et al., 2013*) and animals (*Craft, 2015*; *Rossi et al., 2017*). The network structure exploration is helpful for targeting risk actors and tailoring prevention and control strategies.

Determining a "real" network structure requires knowledge of all individuals in a population and all possible relationships among them. In large networks, this is an impractical and time-consuming task. However, several techniques have been exploited to approximate the structure of the network, for example, a radio-based wearable device was used to identify high-resolution close proximity interactions (less than 1.5 m) among 75 individuals dwelling in 5 different households in rural Kenya (*Kiti et al., 2016*). The study makes it possible to collect a high-resolution human contact data without any direct observations. Similar wireless sensor was also used to explore social contact interactions among students, teachers and staff in an American high school (*Salathé et al., 2010*). The network structure can also be approximated using movement data, for example, airline route maps (*Hufnagel, Brockmann & Geisel, 2004*) or livestock movement patterns (*Wiratsudakul et al., 2014*; *Chintrakulchai, Vuttichai & Wiratsudakul, 2017*; *Khengwa et al.,*

*2017*). However, these data sources have the disadvantage that the network generally links sub-populations or groups of hosts rather than being a network between individuals. Alternatively, the spatial contact proximity can be detected from mobile phones (*Eagle, Pentland & Lazer, 2009*) or other Global Positioning System (GPS) data-loggers (*Vazquez-Prokopec et al., 2013*). Data retrieved from these devices make the contact network to be more realistic which further improve the accuracy of related epidemic models.

Besides using approximated ''real'' networks, several forms of computer-generated networks have also been employed in previous studies. Examples of these ''idealized'' networks include a random network (*Erdös & Rényi, 1959*; *Gilbert, 1959*), in which each pair of nodes is connected randomly, and a scale-free network (*Barabási & Albert, 1999*), where the probability that a node is connected is proportional to its degree. These computer-generated networks are proven to be useful in some aspects of infectious disease transmission (*Keeling & Eames, 2005*; *Pastor-Satorras et al., 2015*). A bipartite network, a network whose nodes are divided into two separate groups with a scale-free degree distribution, was also used to simulate vector-borne disease transmission (*Bisanzio et al., 2010*). The authors found that the spread of disease strongly depends on the degree distribution of the two classes of nodes.

The contact network has also been used to describe disease transmission patterns in mosquito-borne diseases. For instance, a previous study employed a contact-tracing investigation to identify possible contact-site clusters. The authors suggested that house-to-house human movement was likely to indicate how the dengue virus spread spatially (*Stoddard et al., 2013*). This contact-identification technique is applicable to other mosquito-borne diseases, including Zika (*Scatà et al., 2016*; *Saad-Roy, Van den Driessche & Ma, 2016*).

## Individual-based model

The individual-based approach, also known as the agent-based model, allows us to mimic the complexity of individual interactions. Each individual can be explicitly simulated with a set of characteristics including spatial location, interaction preference, behavior traits, etc. Moreover, these state variables dictate how individuals interact with each other. However, they can change over time (*DeAngelis & Grimm, 2014*). Exploitation of the micro-level pattern (the bottom-up method) can prevent the rough estimation that inevitably occurs from the top-down approaches, for example, the compartmental model. The individual-based model is powerful for the integration of different scales and datasets. Therefore, it has been applied to various fields of scientific studies (*El-Sayed et al., 2012*; *Merler et al., 2015*; *Matheson, Satterthwaite & Highlander, 2017*). However, the trade-off between the model complexity and technological requirements must be considered. The realistic models integrating large-scale real-word data apparently demand more sophisticated machines (*Lanzas & Chen, 2015*). Individual-based models have been extensively applied to diseases that require highly unique individual features such as HIV/AIDS (*White et al., 2014*), influenza (*Eichner et al., 2014*), tuberculosis (*Graciani Rodrigues, Espíndola & Penna, 2015*), and Ebola (*Merler et al., 2015*). In mosquito-borne diseases, individual-based models were previously used to describe the transmission dynamics of the chikungunya virus (*Dommar*

*et al., 2014*), the dengue virus (*Chao, Longini & Halloran, 2013*), malaria (*Pizzitutti et al., 2013*), and the Zika virus (*Matheson, Satterthwaite & Highlander, 2017*). It is noteworthy that state-of-the-art structures, including the individual-based, metapopulation, and network models, are not necessarily more realistic than compartmental models. Indeed, the advantage of these modeling structures is that modelers are allowed to fully integrate the models with large-scale real-world data. Consequently, such models are believed to be highly realistic (*Lessler et al., 2016a*; *Lessler, 2016b*). A graphical presentation of the basic models described in this review is illustrated in Fig. 1.

## Mathematical modeling for Zika virus epidemics

The Zika virus has been circulating among human beings for more than 70 years. However, it has been in the sights of modelers for just a decade following a series of outbreaks on Yap Island. Since then, a number of models have been proposed. This study compared some examples based on model structures and discussed the uses of mathematical models in Zika import risk estimation and intervention planning.

## Model architectures

As shown in Table 1, it is noticeable that Zika models carried out for early epidemics were less complicated compared to the more recent ones. To our knowledge, the compartmental approach was a fundamental framework for other sophisticated models that were recently developed. In Zika modeling, all early works were fit into compartmental structures. It was relatively fast and convenient to start with existing knowledge from other related diseases and change the relevant parameters for the Zika virus. However, the compartmental model was still regularly used as a backbone for later models.

Compartmentally, the crisscross transmission between humans and mosquitoes has been popularly simulated. However, the models specially designed for only one (*Monaghan et al., 2016*; *Riou, Poletto & Boëlle, 2017*; *Scatà et al., 2016*) or even another species (*Althouse et al., 2015*) were also observed. Focusing on the model architecture, SEIR was usually used for humans whereas SEI was commonly used for mosquitos. In addition, a model focusing only on human compartments was recently proposed (*Castro et al., 2017*). Nonetheless, other compartmental orientations were occasionally proposed, for example, the susceptible-infectious-recovered (SIR) (*Perkins et al., 2016*), the susceptible-exposed-asymptomatic-infectious-recovered (SEAIR) (*Gao et al., 2016*), the susceptible-preventive isolated-infectious-recovered (S$i^p$IR), and the unaware-aware-faded (UAF) models (*Scatà et al., 2016*).

Spatial models were developed to demonstrate how the Zika virus moves across geographical spaces. Frequently, the spatial framework was complementarily driven by other types of models (*Zinszer et al., 2017*; *Fitzgibbon, Morgan & Webb, 2017*). The most prominent advantage of the spatial models is their visualizing power. Apparently, the maps generated from spatial modeling were the most comprehensible tools for the general public compared to other model outputs. Hence, their final products, viral distribution maps, were frequently exploited in public communication through various channels such as governmental authorities, mainstream media, and even informal online platforms. The

**Table 1  Examples of mathematical models used in Zika virus studies, 2007–2017.** Note that a model is marked as "compartmental" only when the population is divided into groups according to only their health status.

| Period | Location (Country/ Region/Continent) | Population (Compartments) | Model architecture | | | | | References |
|---|---|---|---|---|---|---|---|---|
| | | | Compartmental | Spatial | Metapopulation | Network | Indv.—based | |
| 2007–2012 | Micronesia | Human (SEIR) Mosquito (SEI) | X | | | | | *Funk et al. (2016)* |
| 2007, 2013–2014, 2014 | Micronesia, French Polynesia, New Caledonia | Human (SEIR) Mosquito (SEI) | X | | | | | *Champagne et al. (2016)* |
| 2013–2014 | French Polynesia | Human (SEIR) Mosquito (SEI) | X | | | | | *Kucharski et al. (2016)* |
| 2013–2016 | French Polynesia, French West Indies | Human (SIR) | X | | | | | *Riou, Poletto & Boëlle (2017)* |
| 2014–2017 | American continent | Human (SEIR) Mosquito (SEI) | | | X | | | *Zhang et al. (2017)* |
| 2015 | American continent | Human (SIR) | | X | | | | *Perkins et al. (2016)* |
| 2015–2016 | Brazil | Human (SI) Mosquito (SI) | | X | | | | *Fitzgibbon, Morgan & Webb (2017)* |
| 2015–2016 | Brazil | Human (ND) | | X | | | | *Zinszer et al. (2017)* |
| 2015–2016 | Brazil, Colombia, El Salvador | Human (SEAIR) Mosquito (SEI) | X | | | | | *Gao et al. (2016)* |
| 2016 | United States | Human (SEIR) | X | | | | | *Castro et al. (2017)* |
| 2016 | Brazil | Human (SEIR) Mosquito (SEIR) | | | | | X | *Matheson, Satterthwaite & Highlander (2017)* |
| ND | Brazil | Non-human primates (SIR) Mosquito (SI) | X | | | | | *Althouse et al. (2015)* |
| ND | Worldwide | Human (ND) Mosquito (ND) | | | X | | | *Alaniz, Bacigalupo & Cattan (2017)* |
| ND | ND | Human (SIR/SEIR) Mosquito (SI) | X | | X | | | *Baca-Carrasco & Velasco-Hernández (2016)* |
| ND | ND | Human (SIR) Mosquito (SI) | | | | X | X | *Saad-Roy, Van den Driessche & Ma (2016)* |
| ND | ND | Human (SIR, $Si^pIR$, UAF) | | | | | X | *Scatà et al. (2016)* |

**Notes.**
ND,  Not designated.

disease maps were widely used to increase public awareness and to design specific prevention and control strategies for Zika (*Rodriguez-Morales et al., 2016*) and other emerging diseases (*Coburn & Blower, 2013*; *Emmanuel, Isac & Blanchard, 2013*; *Koch, 2015*).

   We found that the early metapopulation, network, and individual-based models were mostly structured without geographical or timeframe references (Table 1) (*Scatà et al., 2016*; *Baca-Carrasco & Velasco-Hernández, 2016*; *Saad-Roy, Van den Driessche & Ma, 2016*). It seemed difficult to immediately fit the real-world data into these sophisticated

frameworks. However, not long after the Zika epidemic started in Brazil, a data-driven metapopulation model incorporating large-scale real-world data was presented (*Zhang et al., 2017*). The GLEAM model (*Balcan et al., 2010*) was expanded to incorporate data on mosquito density and other entomological-related parameters (*Zhang et al., 2017*). Inclusion of these real-world data into the model is believed to improve the ability of the model to reproduce the observed data and reliably predict future epidemic dynamics.

## Import risk model

Disease transmission models are developed to explore how a pathogen spreads in an epidemic zone. However, the disease, especially a virus, may spread across the globe overnight. Therefore, an import risk model is used in this assessment. A particular framework is designed to quantitatively assess the likelihood of viral importation into a certain territory. Such a model was previously built to evaluate the importation risk of different emerging diseases, for instance, the Ebola virus (*Chen et al., 2014*; *Wiratsudakul et al., 2016*), MERS-CoV (*Nishiura et al., 2015*; *Nah et al., 2016*), and severe acute respiratory syndrome (SARS) (*Goubar et al., 2009*). For the Zika virus, the imported cases were well documented in many countries on different continents (*Pyke et al., 2014*; *Bachiller-Luque, 2016*; *Jang et al., 2016*; *Sokal et al., 2016*; *Zhong et al., 2016*; *Hashimoto et al., 2017*; *Xiang et al., 2017*). The import risk models are necessary to foresee the probability of Zika importation into other unaffected countries. As Brazil was recently in the spotlight for Zika epidemics, models focused on the Zika virus escaping the country were increasingly produced. In particular, models considering the risk of mass gatherings for international events such as the Olympic games were recently proposed (*Grills et al., 2016*; *Massad, Coutinho & Wilder-Smith, 2016*; *Burattini et al., 2016*). Herein, we described three basic methods used in import risk estimation, deterministic and stochastic risk estimation and risk estimation by force of infection.

### Deterministic risk estimation

This method roughly calculated the probability of Zika virus importation into different countries around the world via commercial air travel. In a previous model, the virus was designated to spread from Brazil (*Quam & Wilder-Smith, 2016*). The risk was formulated as $R_I = T \times I \times P$, where the risk ($R_I$) is the product of the number of air passengers ($T$) who traveled from the Zika epidemic areas, the estimated infectious incidence per individual ($I$), and the probability of infection in the travel period ($P$). The results suggested that 584–1,786 Zika cases may have been exported from Brazil during the 2014–2015 epidemics.

### Stochastic risk estimation

This process takes into account the stochasticity of travel volumes. The model was previously employed to estimate the risk of Ebola virus importation into the top 20 destination countries of travelers departing from the three Ebola epidemic countries in West Africa (*Wiratsudakul et al., 2016*). The risk was estimated using the binomial distribution $R_{n,e,t} = Binom(T_{n,e,t} \times I_{n,e,t})$, where $R_{n,e,t}$ represents the risk of viral importation into country $n$ from affected country $e$ at time $t$ whereas $T_{n,e,t}$ and $I_{n,e,t}$ denote the corresponding number of flight travelers and outbreak country incidence,

respectively. This simulation indicated that the risk of importing the Ebola virus during the peak of the epidemics could have reached 0.73 in Ghana, where the highest number of air passengers were observed.

### Risk estimation by force of infection

This method was previously used in the import risk assessment for dengue virus diffusion from Brazil to other countries during the 2016 summer Olympic games (*Ximenes et al., 2016*). First, the force of infection $\lambda$ was estimated from a Gaussian function $\lambda(t) = C_1 \exp\left[-\frac{(t-C_2)^2}{C_3}\right] F(t)$, where $C_1$ determines the highest incidence, $C_2$ is the peak incidence time, and $C_3$ is the period of incidence function. $F(t)$ represents the ad hoc function, which is written in a logistic form as $F(t) = \frac{1}{1+\exp(-C_4(t-C_5))}$, where $C_4$ and $C_5$ are the rate of incidence acceleration and the initial infection time, respectively. Subsequently, $\lambda$ was used to calculate the risk of dengue infection during times $t_1$ and $t_2$ as $\pi(t_1, t_2) = 1 - \exp\left[-\int_{t1}^{t2} \lambda(s)ds\right]$. This model scenario indicated that the number of asymptomatic dengue cases among tourists may have reached 206 during the study period.

## Intervention model

Apart from disease dynamic illustration, mathematical models also functioned as a basic framework to assess the effectiveness of different interventional strategies. For example, a simulated outbreak scenario was examined for the performance of control measures against highly pathogenic avian influenza in Ontario, Canada (*Lewis et al., 2015*). An import risk model was tested for the mitigation capability of pandemic Ebola outbreaks through commercial air travel restrictions (*Wiratsudakul et al., 2016*). The intervention models were also constructed for Zika. A previous study exploited the prior knowledge of rubella control to construct a Zika virus simulation. Rubella is a classic example of teratogenic agents causing viruses in humans. In addition, the body of knowledge on rubella in terms of virology, epidemiology, and mathematical modeling has been well documented (*Metcalf & Barrett, 2016*). This is an excellent example of a modeling framework derived from other well-known diseases. An intervention model for a related emerging disease could be well supported in a timely fashion using such solid mathematical environments.

Theoretical network modeling was also used in the strategic planning for Zika virus outbreak alleviation (*Scatà et al., 2016*). The model selectively removed some specific vertices in the network based on the eigenvector-like centrality and awareness values. Their findings highlighted the importance of heterogeneity and public awareness in the control of infectious diseases under different socioeconomic conditions. Prospectively, the authors planned to include an analogy of HIV epidemics into the sexual transmission of Zika as well as an economic impact evaluation of the disease (*Scatà et al., 2016*). For the economic aspects, a model addressing the cost-effectiveness of Zika control interventions was proposed (*Alfaro-Murillo et al., 2016*). The research team created a user-friendly online tool that was flexible enough to include new parameters and provided a real-time analysis. The program facilitated the financial allocation and assessed its effectiveness. Another example is the economic appraisement of a newly established policy. In the United States, blood centers were ordered to test for the Zika virus to prevent transfusion-transmitted

infection. An economic model was built to assess the implementation costs and to suggest alternatives to reduce them (*Ellingson et al., 2017*). The model was essential for predicting the overall investments and selecting the most cost-effective one. In addition to financial management, other supporting facilities should also be considered. A previous study used a modeling approach to assess the requirement of healthcare resources in real-time (*Andronico et al., 2017*). The authors claimed that their model could provide an accurate prediction.

According to our examples, there are several ways to simulate strategic manipulation using mathematical models. Most modelers designed their models based on existing or newly established policies in order to guide policy makers and precisely meet the needs of societies. However, the field data accuracy and baseline simulations directly affect the prediction power of strategic models. One must seriously consider these factors before translating models into practices.

## Perspectives on Zika virus epidemic models

We noticed that the compartmental model is still commonly used for the timely assessment of epidemics. However, much more complex modeling frameworks (metapopulation, network, and individual-based models) have been of increasing interest due to the recent availability of large-scale real-world data and computational power. As we know, the computational capacity of modern computers is presently very high. This allows us to deploy a sophisticated model to observe the changes and predict the trends of disease dynamics in real-time. Moreover, the model will help policy makers choose the most appropriate intervention to fight outbreaks and further assess the corresponding results in a timely manner. In Zika virus modeling, state-of-the-art structures (the metapopulation, network, and individual-based models) have been increasingly developed using advanced computational capacity. In other diseases such as Ebola, the real-word data was placed into a complicated individual-based modeling framework. The study made it possible to produce a more realistic output reflecting the actual outbreak situations (*Merler et al., 2015*).

"Big data" is an emerging field arising from the extremely large amount of data available together with the advancement in computer infrastructures. In biomedical informatics, large-scale health-related data shared among health professionals are undoubtedly beneficial for the unprecedented development of healthcare services (*Bellazzi, 2014*). Automated modeling could be enabled with the integration of big data and machine learning (*Furqan et al., 2017*). The future of infectious disease modeling including vector-borne diseases may alter the classical methods. Multiple modeling outputs may be generated automatically right after raw data are entered into computers. However, there are some challenges, for example, the reproducibility of the results as well as privacy and data reuse issues (*Bellazzi, 2014*).

In epidemiology, big data are increasingly being used to estimate disease spread and investigate effectiveness of interventions. Recent works in infectious disease dynamics have been characterized by an increasing focus on data-driven approaches. For example, the mobile call data records (CDRs) have been used to explain the dynamics of large-scale Ebola
outbreaks in West Africa (*Wesolowski et al., 2014*). The CDR-based transmission models have also been employed to analyze the spread of rubella disease in Kenya (*Wesolowski et al., 2015*). The individual-based model that integrates detailed geographical and demographic data, and movements of individuals was used to estimate the transmission of Ebola virus and investigate the effectiveness of interventions in Liberia (*Merler et al., 2015*). For Zika, the data-driven metapopulation model integrating real-world demographic, human mobility, socioeconomic, temperature, and vector density data has also been used to analyze the spread of the Zika virus in the Americas (*Zhang et al., 2017*). These emerging data-driven approaches further allow the metapopulation, network, and individual-based models better simulating the real epidemics.

In general, epidemic models can be used either as predictive tools or as a means of understanding fundamental epidemiological processes. However, prediction is perhaps the most obvious use of epidemic models. These allow us to predict the population-level epidemic dynamics from an individual-level knowledge of epidemiological factors, and assess the effectiveness of intervention strategies. As the trends in modeling of infectious diseases have been shifting towards data-driven approaches (*Lessler et al., 2016a*; *Lessler, 2016b*), the model complexity itself may hamper the use of models by nonspecialists and public health practitioners. These complex modeling architectures should be translated into a comprehensible environment. The modelers may adopt some strategies taught in classes on translational medicine to evaluate how to turn epidemic models into practices. Alternatively, user-friendly interfaces are helpful for health professionals to include mathematical models in their strategic plans. For example, the GLEAM framework provides a user-friendly and easy-to-use graphical tool for general modelers and public health agencies (http://www.gleamviz.org). This is an excellent initiation of the translation of complex mathematical models into a touchable framework. The results presented by *Zhang et al. (2017)* were also delivered via user-friendly and easy-to-read graphics on a web application (http://www.zika-model.org/). Therefore, an alliance with computer and graphical scientists is encouraged. Moreover, some educational mobile applications are suggested to acquaint the general public and especially younger generations with epidemic simulations. An excellent example is a simulation game available at the App Store and Google Play called Plague Inc. (Ndemic Creations, Bristol, UK). The game makes mathematical models feel touchable and not too difficult, leading to more familiarity and acceptance.

To fully implement mathematical modeling, one must persuade policy makers to include the methods and try to prove that they are necessary. In this case, the translation of mathematical language into political contexts is crucial. Moreover, simple and complex models should be exploited differently. Simple models can be produced in a timely fashion to provide an estimation of the possible impacts. In contrast, complex models require more time to develop but are far more realistic. The models are much more powerful in terms of predictive capability. The preparation of complicated models before outbreaks is recommended.

## CONCLUSIONS

Mathematical models can be used either as predictive tools or as a means of understanding fundamental epidemiological processes. This review provides basic knowledge of different mathematical models used in studies of disease dynamics. We demonstrated how the models were applied during the course of Zika virus outbreaks and discussed the uses of mathematical models in Zika import risk estimation and intervention planning. We found that Zika models carried out for early epidemics were less complicated compared to the more recent ones. The compartmental model is still commonly used for the timely assessment of epidemics. However, more complex modeling frameworks including metapopulation, network, and individual-based models have been of increasing interest due to the recent availability of large-scale real-world data and computational power. Inclusion of these real-world data into the model is believed to improve the ability of the model to reproduce the observed data and reliably predict future epidemic dynamics.

### Funding

This review study was financially supported by the National Science and Technology Development Agency, Thailand (Project ID P-16-50695) and the Centre of Excellence in Mathematics, the Commission on Higher Education, Thailand. The funders had no role in study design, data collection and analysis, decision to publish, or preparation of the manuscript.

### Grant Disclosures

The following grant information was disclosed by the authors:
National Science and Technology Development Agency, Thailand: Project ID P-16-50695.
Centre of Excellence in Mathematics.
Commission on Higher Education, Thailand.

### Competing Interests

The authors declare there are no competing interests.

### Author Contributions

- Anuwat Wiratsudakul and Charin Modchang conceived and designed the experiments, performed the experiments, analyzed the data, prepared figures and/or tables, authored or reviewed drafts of the paper, approved the final draft.
- Parinya Suparit analyzed the data, approved the final draft.

### Data Availability

    The research in this article did not generate any data or code (literature review).

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
