# Peer review of "Dynamics of Zika virus outbreaks: an overview of mathematical modeling approaches"

_PeerJ, doi:10.7717/peerj.4526_

## Round 0.1 · original submission · Major Revisions

You have reviews from two referees, both of whom feel that revisions are necessary before the journal can accept the paper. One review is anonymous and the other is named. The journal is happy to publish systematic reviews, so you do not need to worry so much about justifying your paper, per se (as one of the referees questioned). However, both of the referees make general and particular points, which you need to somehow address in your revisions if you choose to resubmit.

Reviewer 1 ·

Basic reporting

The manuscript is clearly written in professional English, except some sentences and vocabularies (see comments below). The manuscript contains most of important references in Zika epidemic modeling; however there are still some very significant works missed (see comments below). The structure of this manuscript is very well organized with proper figure and table. Raw data are not applicable for this review manuscript.

Experimental design

The research delivered in this manuscript fits the scopes of the journal. As it is a review paper, methodologies for experimental design are not applicable.

Validity of the findings

I have objections to the conclusion made in this manuscript (see comments below).

Additional comments

As an active researcher in this field, I appreciate the efforts that the authors have made to thoroughly review important works on mathematical modeling for Zika epidemics. However, I have many questions and concerns which needs the authors to address before I can recommend it for publication.

1. The main problem of this manuscript is it lacks of insights. In the Introduction part, the author wrote this "review was aimed to provide a mathematical modeling concept, particularly, of which developed for Zika virus transmission". After I read the article, I did not see any provided concept. What is the concept here? If the concept only means 5 or 6 types of existent models, the authors should be aware that "a concept" does not mean one or two models.

2. The manuscript is nothing but a pile of existent models. The results part is actually not results, but summaries of models which can be found in other references. The manuscript did not see and understand clearly the current trends. For example, the work by Zhang, et al (2017) is NOT a spatial model. It is a meta-population model based on real-world spatial networks using high-resolution spatial environmental data. Thus, what the authors claimed in the perspective paragraphs that (metapopulation, individual-based and network methods) are only a theoretical framework is not true at all. If the authors carefully examine some other papers on agent-based models like "Spatiotemporal spread of the 2014 outbreak of Ebola virus disease in Liberia and the effectiveness of non-pharmaceutical interventions: a computational modelling analysis", they will also find individual-based models or network methods are not theoretical work at all. The recent trends on "big data" allow modeler to have more realistic models. The manuscript would be much more insightful if the authors focus on the latest development on epidemic modelling and the future outlooks.

3. The authors also wrote that there is a lack of user-friendly interface to translate modeling techniques and modeling results. The GLEAM framework in Zhang, et al (2017) has provided a user-friendly and easy-to-use graphical tool for general modelers and public health agencies since years ago, please see www.gleamviz.org. The results presented in Zhang, et al (2017) were also delivered in a user-friendly and easy-to-read graphics on a web application: http://www.zika-model.org/ (this is written in the paper).

4. The manuscript has covered many significant works in mathematical modeling on Zika epidemics. However, there are some very significant papers missed. Below are just a few examples:
Perkins, T. Alex, et al. "Model-based projections of Zika virus infections in childbearing women in the Americas." Nature microbiology 1 (2016): 16126.
Castro, Lauren A., et al. "Assessing real-time Zika risk in the United States." BMC infectious diseases 17.1 (2017): 284.

Obviously, the method to filter literature from google scholar and pubmed is not very well established. The keyword searches should not be too narrowed.

5. The title of this paper is misleading. When I saw the title, I thought this would be a paper representing a new model of zika epidemic dynamics. Please re-title the manuscript.

6. English writing in this manuscript in general is acceptable. However, I recommend the authors to revise the manuscript carefully, especially on vocabularies. For example, in line 64, "jump" does not sound scientifically. What does "jump" mean? Similarly, in line 60, "epical", in line 111, "notorious", etc.

7. In table 1, what are those reference numbers representing?

·

Basic reporting

This literature review article provides a nice introduction to mathematical modeling in the context of Zika. The body of literature that it gathers and synthesizes provides a very useful starting point for anyone wishing to enter this area of research. That said, I think it needs to be improved significantly before it is ready to be published.

Much of the writing is difficult to follow and should be revised and closely edited. The literature is properly referenced in the text, but some important sources appear to be missing (for example Adams and Kapan. 2009. “Man Bites Mosquito: Understanding the Contribution of Human Movement to Vector-Borne Disease Dynamics.” Edited by Alison P. Galvani. PLoS ONE 4 (8).)

The structure is fine, although I recommend making it clearer in the beginning of the Abstract that this is a review article. I am also not sure the authors need to go into such detail about how they selected the articles -- or that they need to limit the literature to that found in this type of methodical survey.

Experimental design

This is not primary research but rather a literature review, summarizing and synthesizing existing work in this area. It appears to fall within the aims and scope of PeerJ. The research question is broad (essentially the article asks what has been done with respect to mathematical modeling relevant to Zika dynamics). It is not entirely clear that it fills a real knowledge gap, although it is surely helpful to have this body of literature synthesized.

The methods in this case are the search criteria. They are clearly described, but I am not convinced that this type of formal survey is needed. I would opt for looser search criteria in order to ensure that all relevant literature has been located. For instance, there are probably important sources that do not contain the term "mathematical model" (even if that is what they provide).

Validity of the findings

The main conclusion is the summary of existing models. From this, the authors note that there has been a lack of application of these models in the field by health professionals, and they suggest some ways in which that might be changed. This is an interesting discussion, but the authors might expand on different ways in which this field implementation could occur, as well as different uses of simple versus complex models.

---

## Round 0.2 · Major Revisions

You have reports from two expert reviewers, one of whom has chosen to make his identity known.

My own read of your revised manuscript was that it did not adopt the suggested revisions from the first round as fully as I would have liked to see.

However, both reviewers feel your revised manuscript was improved. For this reason I am rendering a decision of "major revisions".

Both of the referees make specific and broader recommendations for what they would like to see. If you feel you can implement these changes, then you are invited to re-submit a revised version of the manuscript.

I will not recommend acceptance without support from the referees, who have more specialized subject-matter knowledge on Zika. If you choose to re-submit, then this manuscript will have to be more warmly received by the referees than the current version was.

Reviewer 1 ·

Basic reporting

The English is acceptable. The structure of this manuscript is very well organized with proper figure and table. Raw data are not applicable for this review manuscript.
Experimental design. For comments on technical details, please see the comment section below.

Experimental design

This is a review article, this part is not applicable.

Validity of the findings

See the comments below.

Additional comments

I appreciate the authors' efforts to improve the quality of this survey article. Unfortunately, the authors did not address all of my concerns and questions. There could be some changes in writing and organization of materials to improve the quality.

Line 754, Determining a “real” network structure requires knowledge of all individuals in a population and all possible relationships among them. In large networks, this is an impractical and time-consuming task.

It is true to get real networks, but now there are so many researches using tons of digital proxies to approximately get the human interacting/contacting networks. e.g., "Kiti, M. C., et al. (2016). Quantifying social contacts in a household setting of rural Kenya using wearable proximity sensors. EPJ data science, 5(1), 1-21."

Such digital proxies or data-driven approach allow agent-based models or meta-population models or other models better simulating the real epidemics, and perhaps this is one important reason that there are so many successful papers on Zika modeling since 2016. If the authors go through early publications on modeling Zika epidemics, it can be easily seen that most of early literatures are theoretical and focus on local regions. Until recently, the availability of large-scale real-world data has made computational models approximating the reality, which should be one of the key findings (or punch-line or insight) in this survey article. For example (and once again), the article by Zhang et al, 2017 is NOT A SPATIAL MODEL. It does not fit any part of equation 5. It is a meta-population model on top of human mobility networks. The model includes (nearly) real human mobility data, population data, high-resolution time-varying temperature data, social-economical data and so on.

I would suggest the authors carefully read the literatures covered in the manuscript (e.g., how the complexity of models has evolved and how the application of real-world data has evolved), and revise the manuscript accordingly.

·

Basic reporting

The language and style has improved significantly. I now see only minor improvements to be made here, and these can be mostly handled in the final editing stage. A few specific suggestions:

Lines 560-62: What does this refer back to? I don't see discussion of next-generation matrix approach.
Line 831: I find this sentence awkward.
Line 915: imported risk - import risk
Line 966: A part -> Apart

I am glad to see the Adams and Kapan paper discussed although I think the summary misses the main point of their paper, which is not so much about which places are at risk as about the effects of different patterns of human movement between patches on population-level disease dynamics and outcomes. I would revise this, as it also helps to better explain meta-population theory.

Experimental design

I am glad the authors have checked additional search terms. Perhaps that should be made more clear now in the methods section.

Validity of the findings

The expanded discussion of authors' conclusions is useful. However, this raises and issue I had not fully understood from the previous draft and that I think needs to be addressed: the authors seem to be focused on one type of use for all of these models: making specific geographical predictions about disease spreading. The assumption seems to be that if models cannot be used directly in the field by policy-makers or health professionals to map and predict spreading patterns, then they are not useful. This is what leads to the conclusion about the need for better user interfaces. This leaves out that much of the value of mathematical models is in understanding phenomena in a more abstract way that need not be directly mapped to actual geography (even if this understand does ultimately make it possible to make better predictions in the real world). This need not be done -- in fact, should not be done -- by policy-makers. It should be done by scientists, and lack of user interfaces are not really a barrier.

Similarly, the authors suggest that agent-based models are more realistic than simpler models, but I don't think that is necessarily true. Agent-based models can also be used to better understand mechanisms and other more abstract questions that need not translate directly realistic geographic predictions.

I could see keeping the conclusion about the need for better interfaces to facilitate field implementation as long as the authors make clear that this is only one of many uses of mathematical models. However, I think the conclusion itself also requires more support. It is not clear to me from the literature reviewed why the authors conclude that there is a gap here: The literature discussing the models themselves does not comprehensively assess how these models are used in the field, what is to blame for lack of use, or what we would gain from more field use. I think these points require further analysis.

---

## Round 0.3 · Minor Revisions

Both referees feel you have improved the manuscript. They both have some additional, more minor, suggestions. This is definitely moving in the right direction. I will be pleased to see a revision that addresses the remaining concerns. Thank you for your diligent revisions on prior versions. I am confident you can address the remaining concerns.

Reviewer 1 ·

Basic reporting

English writing is acceptable.
Literature references are sufficient and comprehensive.
The organization of the manuscript has been improved and acceptable.

Experimental design

Not applicable for a review manuscript

Validity of the findings

Not applicable for a review manuscript

Additional comments

Thank all authors for their hard working to improve the manuscript, and most of my concerns in the previous review have been clearly addressed. Below please answer some minor questions before the final acceptance.

Title, I would suggest the title changed from "an overview of a mathematical modeling approach" to "an overview of mathematical modeling approaches"

Line 150, "Our review study was performed from April to July 2017." This sentence is a bit unclear. Does it mean the literature reviewed published in this time period? (which I do not think so). or the activity of searching articles happened in that period. It would be better if just simply saying "we have collected research articled that published before XXXX based on our search criteria"

Line 430, "The most prominent advantage of the spatial models is their virtualizing power." What does the term "virtualizing" mean here?

Regarding to "epidemic management" or intervention, in the scenario of Zika epidemics, the authors could consider this working paper to gain some insights https://www.biorxiv.org/content/early/2017/09/18/187591

·

Basic reporting

All of my previous suggestions have been addressed.

Experimental design

All of my previous suggestions have been addressed.

Validity of the findings

All of my previous suggestions have been addressed.

Additional comments

The authors have done a great job of addressing my previous suggestions. I have attached a marked-up PDF with some very minor additional edits.

---

## Round 0.4 · accepted · Accept

You have now satisfied both referees, and me. I'm pleased to accept your submission.

Reviewer 1 ·

Basic reporting

English writing is acceptable.

Literature references are sufficient and comprehensive.

The organization of the manuscript is well structured.

Experimental design

NA

Validity of the findings

NA

Additional comments

Thank all authors for their efforts in improving the quality of this manuscript. The authors have addressed all my questions.